# Diet and Lifestyle Factors and Risk of Atherosclerotic Cardiovascular Disease—A Prospective Cohort Study

**DOI:** 10.3390/nu13113822

**Published:** 2021-10-27

**Authors:** Stefan Acosta, Anna Johansson, Isabel Drake

**Affiliations:** 1Department of Clinical Sciences, Lund University, 214 21 Malmö, Sweden; anna.johansson.5408@med.lu.se (A.J.); isabel.drake@med.lu.se (I.D.); 2Vascular Center, Department of Cardiothoracic and Vascular Surgery, Skåne University Hospital, 205 01 Malmö, Sweden; 3Örestadskliniken, 217 67 Malmö, Sweden

**Keywords:** atherosclerosis, cardiovascular disease, epidemiology, diet quality, fish, fiber, saturated fatty acids, physical activity, lifestyle

## Abstract

Atherosclerotic cardiovascular disease (ACVD) is the leading cause of death worldwide. This study aimed to investigate the association between diet and lifestyle factors, beyond traditional risk factors, and the risk of incident ACVD. The Malmö Diet and Cancer study included 30,446 middle-aged individuals. Baseline examinations including a dietary assessment, questionnaire and interviews, were performed between 1991–1996. After excluding individuals with prevalent cardiovascular disease and atrial fibrillation or flutter, 26,990 participants remained. In a previously developed diet quality index, adherence to recommended intake of saturated fat (SFA), polyunsaturated fat (PUFA), fish and shellfish, fiber, vegetables and fruit, and sucrose results in one point per dietary component, with a maximum diet score of six points. Diagnosis of incident ACVD was based on validated diagnoses of coronary artery disease, atherothrombotic ischemic stroke, carotid artery disease or peripheral artery disease. Multivariable Cox regression analysis adjusting for established risk factors was performed to assess hazard ratios (HR) with 95% confidence intervals (CI). After a median follow-up of 21.1 years, 5858 (21.7%) individuals diagnosed with ACVD unrelated to atrial fibrillation or flutter were identified. Higher diet score (HR 0.94/point increase; 95% CI 0.91–0.97; *p* < 0.001), intake of fish and shellfish (HR 0.95/standard deviation (SD) increment, 95% CI 0.93–0.98), fiber (HR 0.93/SD increment, 95% CI 0.89–0.98) and SFA (HR 0.96/SD increment, 95% CI 0.92–0.99) consumption were associated with decreased risk for incident ACVD. High leisure-time physical activity (HR 0.82, 95% CI 0.74–0.91) was associated with reduced risk and obesity (HR 1.17, 95% CI 1.08–1.27) with increased risk of incident ACVD. The present study strengthens current recommendations of improving diet quality and increasing physical activity in preventing ACVD.

## 1. Introduction

Atherosclerotic cardiovascular disease (ACVD) of the coronary and cerebral arteries are the two leading causes of death in the world [1]. Over 17 million individuals died from cardiovascular disease (CVD) in 2015, representing 31% of all global deaths. Globally, >75% of deaths from CVD occur in low- and middle-income countries. In these countries, individuals with CVD have limited access to effective and equitable healthcare services, resulting in delays in CVD diagnosis and treatment and premature mortality.

Some of the traditional non-modifiable risk factors for atherosclerotic disease are age and male sex; major semi-modifiable risk factors include hypertension, diabetes mellitus (DM) and high cholesterol. Despite great efforts to reduce ACVD burden with conventional risk factor control such as smoking cessation, control of systolic blood pressure and total cholesterol, the worldwide rise in obesity and diabetes mellitus threatens to counteract these preventive efforts [2]. From a global health and economic perspective, effective and durable lifestyle interventions are most wanted for the primary prevention of ACVD.

Smoking, physical activity, diet and weight are the four core health behaviors [3,4] that contribute to arterial health. A high diet quality was associated with lower incidence of cardiovascular events in terms of myocardial infarction or ischemic stroke with follow up to 2008 in the Malmö Diet and Cancer Study (MDCS) [5]. A healthy diet and adherence to recommended levels of fiber intake were associated with a reduced risk of incident peripheral artery disease in MDCS [6], whereas no association was found between diet quality and incident carotid artery disease [7], but this study was probably underpowered. Current smoking, low diet quality and low physical activity level were associated with atherothrombotic ischemic stroke in MDCS [8]. However, data on the influence of all four core modifiable risk factors on incident ACVD, i.e., either coronary artery disease, atherothrombotic ischemic stroke, carotid artery disease or peripheral artery disease, have been sparse and such evaluation has not been conducted in the MDCS cohort.

The purpose of the present population-based prospective study was to evaluate the association between the four core lifestyle factors and risk of incident ACVD.

## 2. Materials and Methods

### 2.1. Study Population and Data Collection

Baseline examinations were carried out between 1991 and 1996 in men and women aged 46–73 that were living in Malmö at the time of enrolment [9]. The cohort was followed until end of 2016. Out of the 30,446 participants included, 28,098 participants underwent anthropometric measurements and a dietary assessment. Participants with prevalent atrial flutter or fibrillation (AF), coronary artery disease, ischemic stroke, carotid artery disease and peripheral artery disease were excluded, after which 26,990 remained to be included in the present study (Figure 1). The study was conducted in accordance with the World Medical Association Declaration of Helsinki and the study protocol was approved by the Ethical Committee at Lund University (LU 51-90) and by the Regional Ethical Review Board in Lund, Sweden (Dnr 2007/166). All subjects gave their written consent for participation in the study.

### 2.2. Diet Assessment Method and Diet Quality Components

The participants filled in a 7-day food diary and a 168-item food frequency questionnaire that included foods regularly consumed the past year. Complementary information was gathered through 1-h interviews. A diet quality index based on the Swedish nutrition recommendations [10] has been developed and validated within the MDCS cohort [11]. The index includes intake of six dietary components: saturated fatty acids (SFA) ≤ 14 energy (E)%, polyunsaturated fatty acids (PUFA) 5–10 E%, fish and shellfish ≥ 300 g/week, sucrose ≤ 10 E%, dietary fiber ≥ 2.4 g/megajoule (MJ), fruit and vegetables ≥ 400 g/day. A reached recommendation results in one point per dietary component, with a maximum score of six points. A low score (0–1 points) indicates low quality diet, medium score (2–4 points) medium quality diet and high score (5–6 points) high quality diet [11].

### 2.3. Lifestyle and Other Variables

Data on smoking, alcohol habits, education level and leisure-time physical activity level was collected via questionnaires. Smoking was defined as never, former or current smoking. Alcohol intake registered in the 7-day food diary was categorized according to gender specific alcohol limits (quintiles) and zero-consumers were defined as individuals who had not consumed alcohol the past year based on their report in the baseline questionnaire. Leisure-time physical activity level was measured in metabolic equivalent of task hours per week, defined by intensity level and time spent on 17 different activities, adapted from the Minnesota Leisure Time Physical Activity Questionnaire [12]. Body mass index (BMI) < 25 kg/m^2^, 25–29.99 kg/m^2^ and ≥30 kg/m^2^ defined normal weight, overweight and obesity, respectively. Hypertension was defined as systolic blood pressure ≥ 140 mmHg, diastolic blood pressure ≥ 90 mmHg or current use of antihypertensive medication. DM was defined as having a measured fasting whole blood glucose ≥ 6.1 mmol/L, self-reported history of physician-diagnosed diabetes, use of diabetes medication, or being diagnosed and registered in any of the local or national diabetes registries. Attained education level was defined as less than 9 years, elementary school (9–10 years), upper secondary school (11–13 years), university without a degree, and university degree.

### 2.4. Endpoint Ascertainment

Incident ACVD was defined as diagnosis of coronary artery disease, atherothrombotic ischemic stroke, carotid artery disease or peripheral artery disease [13]. The Swedish National Patient register and the Cause of Death Register Participants were used to identify participants with a first registered diagnosis of ACVD via civic registration numbers. Diagnoses are coded using a Swedish revision of the International Classification of Disease (ICD), versions 8, 9 and 10. Patients registered with atrial fibrillation or flutter (AF) prior to or simultaneously (±30 days) to ischemic stroke were labeled as AF related ischemic stroke and were excluded as atherothrombotic ischemic stroke. AF related ischemic stroke patients were followed up until date of incident AF. AF was ascertained by ICD8-427.9, ICD9-427D and ICD10-I48 codes. Follow up time was determined by date of first incident ACVD, death or end of follow up. The diagnoses of coronary artery disease, ischemic stroke, carotid artery disease and peripheral artery disease in the Swedish National Patient register were separately validated by selecting a random sample of 100 study participants for each diagnosis (Appendix A: Validation of diagnosis of atherosclerotic cardiovascular disease). During follow up, 6339 participants in the final study population were diagnosed with any incident ischemic cardiovascular disease (coronary artery disease, all-cause ischemic stroke, carotid artery disease or peripheral arterial disease) of which 5858 (92.4%) were classified as ACVD (excluding AF-related embolic strokes).

### 2.5. Statistical Analysis

Baseline characteristics were expressed as median and interquartile range (IQR) for continuous variables, and as total count and percentage for categorical variables. The proportional hazards assumption was tested by stratifying by examined variables. The plots of the estimated log-log survival curves were found to be approximately parallel and fulfilled the proportional hazards assumption. Multivariable Cox proportional hazards regression analysis was used to calculate hazard ratios (HR) with 95% CI with mutual adjustment for the included risk factors. Dietary components, total energy intake, age and BMI were tested for normal distribution with the Kolmogorov-Smirnov test. All these variables were log transformed due to skewed distribution, and HRs were expressed per 1 standard deviation (SD) increment. For statistical analyses IBM SPSS Statistics, version 26 (SPSS, Chicago, IL, USA) was used, and level of statistical significance was *p* < 0.05.

## 3. Results

### 3.1. Baseline Characteristics

After a median of 21.1 years (IQR 15.0–23.1) of follow-up, 5858 (21.7%) out of 26,990 patients were diagnosed with atherosclerotic disease. Among the 5858 patients, the first ACVD events were caused by coronary artery disease (*n* = 2616; 44.7%), atherothrombotic ischemic stroke (*n* = 1976; 33.7%), carotid artery disease (*n* = 384; 6.6%) and peripheral artery disease 882 (15.1%) (Figure 2). Baseline characteristics for those with or without incident ACVD are presented in Table 1. Age (HR 1.78 per SD increment, 95% CI 1.76–1.86; *p* < 0.001) and male sex (HR 1.79, 95% CI 1.68–1.90; *p* < 0.001) were associated with higher risk and having a university degree compared to not having completed elementary school (HR 0.74, 95% CI 0.67–0.82; *p* < 0.001) with lower risk of incident ACVD. Hypertension (HR 1.50, 95% CI 1.41–1.60; *p* < 0.001) and diabetes mellitus (HR 2.26, 95% CI 2.05–2.49; *p* < 0.001) at baseline were associated with increased risk of incident ACVD.

### 3.2. Diet Quality Index Components and Atherosclerotic Disease Risk

Intake of fish and shellfish (HR 0.95 per SD increment, 95% CI 0.93–0.98; *p* = 0.001), dietary fiber (HR 0.93 per SD increment, 95% CI 0.89–0.98; *p* = 0.004) and SFA (HR 0.96 per SD increment, 95% CI 0.92–0.99; *p* = 0.020) was associated with reduced risk of ACVD in the multivariable Cox regression analysis including mutual adjustment for the six diet quality index components. When entering the diet score instead of intake of the diet quality index components in the multivariable Cox regression model, diet score (HR 0.94 per point increase; 95% CI 0.91–0.97; *p* < 0.001) was associated with decreased risk for incident ACVD (Table 1). There was no association between adherence (yes/no) to the diet quality index components and risk for ACVD (Appendix A: HR and 95% CI for incident atherosclerotic cardiovascular disease by adherence to diet quality index components).

### 3.3. Other Lifestyle Factors and Atherosclerotic Disease Risk

Current smokers as opposed to never smokers had an increased risk of incident ACVD (HR 2.24, 95% CI 2.10–2.40; *p* < 0.001). Medium (HR 0.87, 95% 0.79–0.94; *p* = 0.001) and high (HR 0.88, 95% CI 0.80–0.96; *p* = 0.006) alcohol consumers as opposed to low alcohol consumers had a reduced risk of incident ACVD. Medium (HR 0.77, 95% CI 0.70–0.85; *p* < 0.001) and high (HR 0.82, 95% CI 0.74–0.91; *p* < 0.001) leisure-time physical activity as opposed to low physical activity were associated with reduced risk of incident ACVD. Higher BMI (HR 1.06 per SD increment, 95% CI 1.03–1.09; *p* < 0.001) was associated with increased risk of incident ACVD (Table 1). When instead entering BMI as a categorical variable in the multivariable Cox regression analysis, obesity (HR 1.17, 95% CI 1.08–1.27; *p* < 0.001) and overweight (HR 1.09, 95% CI 1.03–1.16; *p* = 0.004) as opposed to normal weight were associated with increased risk of incident ACVD.

## 4. Discussion

This population-based prospective cohort study with a median follow-up of 21.1 years found, beyond traditional risk factors, an association between a healthy lifestyle and reduced risk of ACVD. Higher overall diet quality, higher fish and shellfish, fiber and SFA intake, and higher physical activity level, reduced the risk of ACVD, whereas smoking and elevated BMI increased the risk of ACVD. Higher as opposed to lower alcohol consumption was also associated with reduced risk of ACVD.

The importance of lifestyle factors beyond traditional risk factors for the prevention of ACVD is receiving increasing attention [14]. Evidence from randomized trials on cardiovascular endpoints provides robust support for the superiority of food-based recommendations without specific guidance regarding calories, as opposed to nutrient- and calorie-based recommendations [15]. The Mediterranean diet pattern, distinguished as high consumption of plant-based foods, olive oil as the main source of fat, moderate consumption of fish, dairy products and poultry, low consumption of red and processed meat, and low-to-moderate consumption of wine with meals, supplemented by extra virgin olive oil or nuts, is the most established dietary pattern to reduce cardiovascular endpoints [16,17] and dyslipidemia [18]. In the present study a high intake of fish and shellfish was found to be a protective factor for ACVD. This result is in line with a prospective longitudinal study in individuals with diabetes mellitus [19] and the positive effects ascribed to the Mediterranean diet pattern. A healthy diet, rich in fish, might help to achieve and maintain body weight goals, reach individual glycemic, blood pressure, and lipid targets, and prevent diabetic complications [20].

The association between fiber intake and reduced risk of incident peripheral atherosclerotic occlusive disease [6] was further strengthened by the present study finding on the composite endpoint ACVD. Soluble fiber prolongs the emptying of the stomach and food transit time by forming a gel, lowering postprandial blood glucose and lipid increase [21]. Fiber also distends the stomach, resulting in hormone secretion that increases satiation, thereby reducing food intake and improving glucose metabolism. By decreasing body weight, fiber intake also reduces blood pressure. In addition, the rate of bile acid excretion has been shown to be increased by dietary fiber, lowering total and low-density lipoprotein cholesterol. Cholesterol synthesis is also inhibited by production of short-chain fatty acids from fiber by bacteria in the gut. Hence, the protective effects of fiber may be mediated by reducing hyperglycemia, hyperlipidemia, and hypertension [22,23]. It is also possible that high consumers of fish and shellfish and fiber have a different lifestyle compared to low consumers, which might help explain the putative protective effects towards development of ACVD.

SFA was associated with reduced risk of ACVD, which has previously been shown for coronary events in women in the MDCS cohort [24]. This replicated finding might at first impression seem to be controversial. First, one should be aware that the major sources of SFAs include a wide variety of foods, such as milk, cheese, butter, margarine, plant oils, red and processed meats, poultry, chocolate, and baked desserts. Intake of SFAs derived from meats have been linked to higher risk of CVD, whereas intake of SFAs derived from dairy have been linked to lower risk [25]. Of note, a cross sectional analysis of food sources in MDCS participants showed that dairy products and margarine contributed more than twice as much to total fat and energy intake compared to meat products in both women and men [26]. Secondly, consumption of processed meats as opposed to unprocessed meats is associated with an elevated risk of CVD [27,28,29], which may be attributed by high sodium content and harms of other preservatives such as nitrites in processed meats [30]. Thirdly, different medium-chain and long-chain SFAs have very different effects on blood lipids [31]. Further research into the mechanistic and health effects of individual fatty acids and their different food sources will be important.

Medium and high levels of leisure-time physical activity compared to low level of physical activity were associated with decreased risk of ACVD in the present study. Physical inactivity and low cardiorespiratory fitness are under-recognized cardiovascular risk factors and are strong independent predictors of outcomes in primary and secondary prevention of ACVD across different BMI groups [32,33]. The protective effect of physical activity on ACVD could be explained by its ability to reduce blood pressure, blood lipids, excess body weight and preventing diabetes mellitus [34]. BMI can be seen as an intermediary variable and excess body weight was indeed found to be associated with increased ACVD risk in the present study. This result suggests that reduction of weight through healthy dietary habits and physical activity may prevent ACVD.

Medium and high levels of alcohol consumption were associated with reduced ACVD risk in the present study. Belonging to the highest category of alcohol drinkers means that you consume slightly more than one glass of wine (12 cl) per day if you are a woman or slightly more than two glass of wine per day if you are a man. Since the participation rate of the study was 40% and participants reported better health than non-participants [35], one might suspect that the amount of alcohol consumption among study participants was generally low, resulting in low limits for every category of alcohol consumption. Hence, misclassification of alcohol intake might affect observed associations, and a greater percentage of heavy alcohol drinkers were probably to be found among the non-participants.

The limitation of the present study includes the risk of misreporting of self-reported dietary and lifestyle habits. In addition, assessment of lifestyle risk factors was carried out at baseline only. Iteration of these measurements would lower the risk of potential residual confounding. The main strengths include the large study population, the large number of ACVD events by using a composite endpoint of the most common atherosclerotic diseases and a follow-up time of over 20 years. Most previous studies on diet and lifestyle in relation to cardiovascular disease have examined a composite of coronary artery disease and all-cause ischemic strokes, or these two endpoints separately. The advantage of the current study was the examination of risk factors for ACVD, excluding ischemic strokes likely to be caused by AF-related cardiac embolization and including less commonly studied atherosclerotic endpoints such as carotid artery disease and peripheral artery disease. The fact that registries were used to identify endpoints ensured nearly no loss to follow up. The diet quality index used has previously been shown to have high content validity for assessing adherence to the Swedish nutrition recommendations [11]. Furthermore, the extensive validation of endpoint diagnosis showed high validity for the diagnosis coronary artery disease, carotid artery disease and peripheral artery disease, while validation of participants with incident ischemic stroke showed that approximately one-third of the ischemic strokes were caused by embolization secondary to AF and not by atherothrombotic ischemic stroke. Therefore, adjustments for ischemic stroke unrelated to AF were performed in the dataset to only include participants with atherosclerotic thrombotic events. The ascertainment of AF in the MCDS cohort has previously been shown to have high validity after scrutinizing electrocardiograms [36].

## 5. Conclusions

In conclusion, the present study strengthens current recommendations of improving diet quality and increasing physical activity in preventing ACVD.

## Figures and Tables

**Figure 1 nutrients-13-03822-f001:**
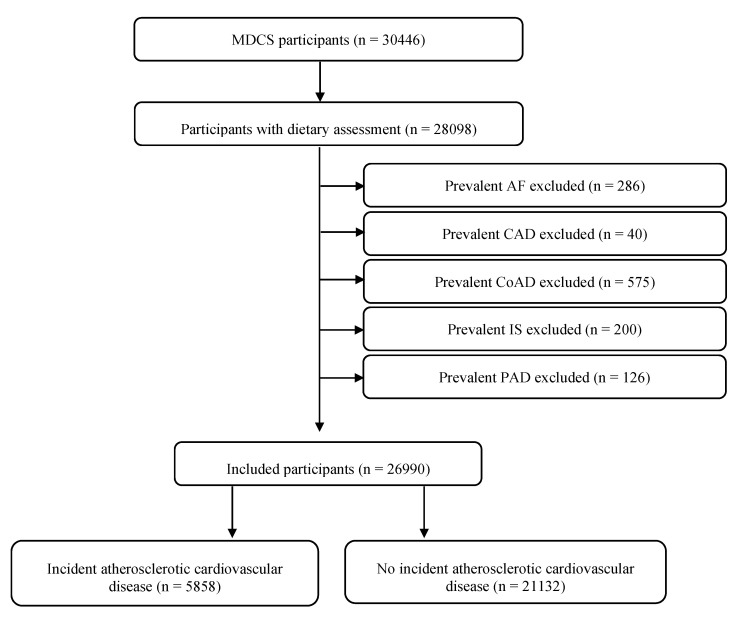
Descriptive flow diagram of study participants, dietary data and exclusions. MDCS: Malmö Diet and Cancer Study; AF: atrial flutter or fibrillation; CAD: carotid artery disease; CoAD: coronary artery disease; IS: ischemic stroke; PAD: peripheral artery disease. Some individuals had multiple exclusion criteria.

**Figure 2 nutrients-13-03822-f002:**
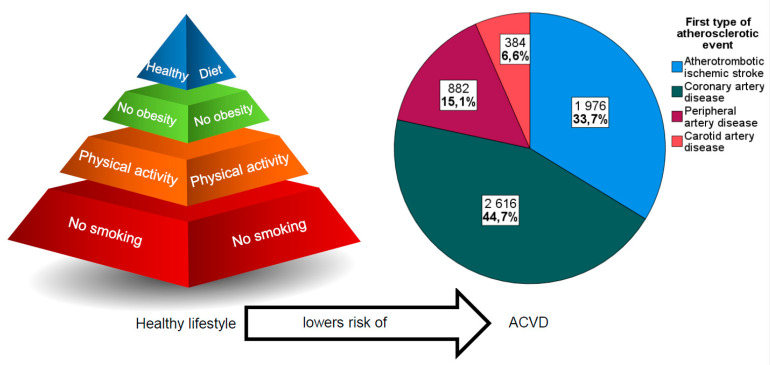
Graphical schematic presentation of the main results of the study, where no smoking has the largest reduction in risk of the modifiable risk factors for atherosclerotic cardiovascular disease (ACVD).

**Table 1 nutrients-13-03822-t001:** Baseline characteristics of study participants with and without incident atherosclerotic cardiovascular disease in the MDCS cohort.

	Incident Atherosclerotic Cardiovascular Disease (*n* = 5858)	No Incident Atherosclerotic Cardiovascular Disease (*n* = 21,132)	Age-and Sex Adjusted HR (95% CI)	Multivariable * Adjusted HR (95% CI)
Male sex (%)	2991 (51.1)	7290 (34.5)	1.85 (1.75–1.95)	1.79 (1.68–1.90)
Age (years)	61.8 (55.5–66.2)	56.1 (50.3–63.0)	1.81 ^a^ (1.76–1.86)	1.78 ^a^ (1.72–1.84)
Total energy intake (kcal/day)	2223.1 (1840.4–2703.9)	2174.6 (1815.0–2611.8)	0.95 ^a^ (0.92–0.98)	0.99 ^a^ (0.96–1.02)
BMI (kg/m^2^)	25.9 (23.5–28.4)	25.1 (22.8–27.7)	1.10 ^a^ (1.07–1.13)	1.06 ^a^ (1.03–1.10)
Hypertension (%)	4318/5842 (73.9)	11,983/21,092 (56.8)	1.52 (1.43–1.62)	1.50 (1.41–1.60)
Diabetes mellitus (%)	480 (8.2)	607 (2.9)	2.36 (2.15–2.60)	2.26 (2.05–2.49)
Alcohol Consumption (%)				
Zero-consumers	455 (7.8)	1261 (6.0)	1.15 (1.04–1.29)	1.10 (0.98–1.23)
Quintile 1 (<0.9 g/day for women/<3.4 g/day for men)	1234 (21.1)	3766 (17.8)	1 (Ref)	1 (Ref)
Quintile 2 (0.9–4.3 g/day for women/3.4–9.1 g/day for men)	1181 (20.2)	3852 (18.2)	0.91 (0.84–0.98)	0.95 (0.87–1.03)
Quintile 3 (4.4–8.1 g/day for women/9.2–15.7 g/day for men)	1056 (18.0)	4023 (19.0)	0.80 (0.74–0.87)	0.87 (0.79–0.94)
Quintile 4 (8.2–14.0 g/day for women/15.7–25.7 g/day for men)	973 (16.6)	4095 (19.4)	0.78 (0.72–0.85)	0.84 (0.77–0.92)
Quintile 5 (>14.0 g/day for women/>25.7 g/day for men)	959 (16.4)	4135 (19.6)	0.86 (0.79–0.94)	0.88 (0.80–0.96)
Smoking (%)				
Never	1886/5853 (32.2)	8519/21,125 (40.3)	1 (Ref)	1 (Ref)
Former	1882/5853 (32.2)	7057/21,125 (33.4)	1.17 (1.09–1.25)	1.18 (1.10–1.26)
Current	2085/5853 (35.6)	5549/21,125 (26.3)	2.16 (2.02–2.31)	2.24 (2.10–2.40)
Leisure-Time Physical Activity (%)				
<7.5 MET-h/week	667/5803 (11.5)	1917/20,998 (9.1)	1 (Ref)	1 (Ref)
7.5–15.0 MET-h/week	935/5803 (16.1)	3068/20,998 (14.6)	0.84 (0.76–0.93)	0.92 (0.83–1.02)
15.1–25.0 MET-h/week	1222/5803 (21.3)	4941/20,998 (23.5)	0.68 (0.62–0.75)	0.77 (0.70–0.85)
25.1–50.0 MET-h/week	1992/5803 (21.1)	7758/20,998 (36.9)	0.69 (0.63–0.75)	0.81 (0.73–0.88)
>50.0 MET-h/week	987/5803 (17.0)	3314/20,998 (15.8)	0.71 (0.64–0.79)	0.82 (0.74–0.91)
Educational Level (%)				
Less than 9 years	3020/5840 (51.7)	8143/21,088 (38.6)	1 (Ref)	1 (Ref)
Elementary school (9–10 year)	1379/5840 (23.6)	5728/21,088 (27.2)	0.83 (0.78–0.89)	0.89 (0.84–0.96)
Elementary + upper secondary school (9–13 year)	469/5840 (8.0)	1937/21,088 (9.2)	0.77 (0.70–0.86)	0.86 (0.78–0.96)
University studies, no degree	418/5840 (7.2)	1940/21,088 (9.2)	0.73 (0.65–0.81)	0.82 (0.74–0.92)
University studies, with degree	554/5840 (9.5)	3340/21,088 (15.8)	0.63 (0.57–0.69)	0.74 (0.67–0.82)
Diet Quality				
Low (%)	962 (16.4)	3202 (15.2)	1 (Ref)	1 (Ref)
Medium (%)	4134 (70.6)	15,114 (71.5)	0.85 (0.80–0.92)	0.91 (0.84–0.98)
High (%)	762 (13.0)	2816 (13.3)	0.82 (0.75–0.91)	0.92 (0.83–1.01)
Diet score (0–6)	3 (2–4)	3 (2–4)	0.95 (0.92–0.98)/ point increase	0.94 (0.91–0.97)/ point increase
Dietary Components				
Saturated fat (E%)	15.7 (13.6–18.5)	15.8 (13.7)	0.93 ^a^ (0.89–0.96)	0.96 ^a^ (0.92–0.99)
Polyunsaturated fat (E%)	5.8 (4.9–6.9)	5.7 (4.8–6.8)	1.02 ^a^ (0.99–1.05)	1.02 ^a^ (0.99–1.05)
Sucrose (E%)	8.0 (5.9–10.4)	8.0 (6.1–10.3)	0.96 ^a^ (0.93–0.99)	1.01 ^a^ (0.98–1.04)
Fiber (g/MJ)	2.1 (1.7–2.5)	2.1 (1.8–2.6)	0.88 ^a^ (0.84–0.93)	0.93 ^a^ (0.89–0.98)
Vegetables and fruit (g/day)	333.6 (229.5–455.6)	351.5 (250.5–479.8)	0.96 ^a^ (0.92–0.99)	0.99 ^a^ (0.95–1.03)
Fish (g/week)	284.4 (151.8–451.6)	274.6 (148.4–434.3)	0.93 ^a^ (0.91–0.96)	0.95 ^a^ (0.93–0.98)

Data are *n* (%), or median (IQR). BMI, body mass index; E, energy; MET, metabolic equivalent of task; MJ, megajoule. ^a^ HR were expressed per 1 SD increment. * Multivariable model includes all risk factors and diet quality respective dietary component variables. Dietary component variables were mutually adjusted.

## Data Availability

Restrictions apply to the availability of these data. Data was obtained from The Malmö Cohorts and are available at https://www.malmo-kohorter.lu.se/malmo-cohorts (accessed on 1 October 2021) with the permission of MDC Steering Committee.

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
