# Peer review of "Diet and Lifestyle Factors and Risk of Atherosclerotic Cardiovascular Disease—A Prospective Cohort Study"

_nutrients, 2021, doi:10.3390/nu13113822_

Round 1

Reviewer 1 Report

The paper is well written. The data seems to be similar to previous literatures as the authors claim in “conclusion”. What extra did this study provide to the existing literature must be written, or the expectation should be mentioned.

Incident ACVD was defined as diagnosis of coronary artery disease, atherothrombotic ischemic stroke, carotid artery disease or peripheral artery disease.

Add reference

Soluble fiber prolongs the emptying of the stomach by  forming a gel, which is thought to lower postprandial blood glucose and lipid increase.

Add reference

Table 1 contain a panel that states “Dietary components” as polyunsaturated fats.

Please mention the ώ6 content as well as ώ3 content of the fats.

Similarly, fiber should be categorized into soluble and insoluble fiber

In the conclusion the authors state” In conclusion, the present study strengthens current recommendations of improving diet quality and increasing physical activity in preventing ACVD”.

What was the expectation of the authors when they started this study? Is the study just a validation?

Author Response

The paper is well written. The data seems to be similar to previous literatures as the authors claim in “conclusion”. What extra did this study provide to the existing literature must be written, or the expectation should be mentioned.

Incident ACVD was defined as diagnosis of coronary artery disease, atherothrombotic ischemic stroke, carotid artery disease or peripheral artery disease.

Add reference

Answer: This is a composite variable of the four most common atherosclerotic diseases. We add a new reference, reference 13, in Methods - Endpoint ascertainment: Tran H, Anand SS. Oral antiplatelet therapy in cerebrovascular disease, coronary artery disease and peripheral arterial disease. JAMA 2004; 292: 1867 – 74.

Soluble fiber prolongs the emptying of the stomach by  forming a gel, which is thought to lower postprandial blood glucose and lipid increase.

Add reference

We add new reference 21: Soliman GA. Dietary fiber, atherosclerosis, and cardiovascular disease. Nutrients 2019; 11:1155. Doi: 10.3390/nu11051155.

Table 1 contain a panel that states “Dietary components” as polyunsaturated fats.

Please mention the ώ6 content as well as ώ3 content of the fats.

As we only studied the dietary components that where included in the diet quality index, we think it is out of scope to include that data in the paper.

Similarly, fiber should be categorized into soluble and insoluble fiber

We have no data in the database that are categorized as soluble and insoluble fiber.

In the conclusion the authors state” In conclusion, the present study strengthens current recommendations of improving diet quality and increasing physical activity in preventing ACVD”.

What was the expectation of the authors when they started this study? Is the study just a validation?

No. Healthy diet has previously been shown to decrease risk of peripheral arterial disease in the present cohort, but not for physical activity. By increasing the number of individuals with atherosclerotic disease it was possible to show several additional significant findings, including physical activity. Previous studies on cardiovascular disease have not had a refined atherosclerotic disease endpoint as pointed out in the abstract and manuscript of the present study. This has been elaborated upon in the limitation of the study section in Discussion.

Reviewer 2 Report

  • please discuss the role of Mediterranean diet on such a context. Please consider and discuss such a point in relation to the paper from Mattioli AV ert al. J Cardiovasc Med (Hagerstown). 2017 Dec;18(12):925-935 and Nutr Metab Cardiovasc Dis. 2017 Nov;27(11):985-990.
  • - please include a representative figure for this paper in order to improve the readability of the text.

Author Response

please discuss the role of Mediterranean diet on such a context. Please consider and discuss such a point in relation to the paper from Mattioli AV ert al. J Cardiovasc Med (Hagerstown). 2017 Dec;18(12):925-935 and Nutr Metab Cardiovasc Dis. 2017 Nov;27(11):985-990.

Thanks, we added this reference as new reference 17, when we discussed the dietary patterns to reduce cardiovascular endpoints.

- please include a representative figure for this paper in order to improve the readability of the text.

Please see the attachment. I could not download the complementary figure legend in word format.

We inserted a new representative figure for the main results of this paper as suggested, figure 2, with the figure legend: "Graphical schematic presentation of the main results of the study, where no smoking has the largest reduction in risk of the modifiable risk factors for atherosclerotic cardiovascular disease (ACVD)."

We refer to Figure 2 in the first paragraph of the Results – Baseline characteristics and first paragraph in the Discussion
